# Glycerol Carbonate Solventless Synthesis Using Ethylene Carbonate, Glycerol and a Tunisian Smectite Clay: Activity, Stability and Kinetic Studies

**Yosra Snoussi** [1,2], **Itziar A. Escanciano** [3], **Mariana Alvarez Serafini** [4], **Neji Besbes** [1,2], **Juan M. Bolivar** [3] and **Miguel Ladero** [3,*]

1   Laboratoire Matériaux Composites et Minéraux Argileux, Centre National de Recherches en Sciences des Matériaux, Technopole Borj Cédria, Soliman 8027, Tunisia; yosrasnoussi2016@gmail.com (Y.S.); besbesneji@yahoo.fr (N.B.)
2   Tunisie Faculté des Sciences Mathématiques, Physiques et Naturelles de Tunis, Tunis 2092, Tunisia
3   FQPIMA Group, Materials and Chemical Engineering Department, Chemical Science School, Complutense University of Madrid, 28040 Madrid, Spain; itziaria@ucm.es (I.A.E.); juanmbol@ucm.es (J.M.B.)
4   Chemical Engineering Department, Universidad Nacional del Sur (UNS) and Pilot Plant-PLAPIQUI (UNS-CONICET), Bahía Blanca B8000, Argentina; alvarezserafini@gmail.com
*   Correspondence: mladerog@ucm.es; Tel.: +34-91-394-4164

**Featured Application: A natural Tunisian carbonate-rich clay is an active catalyst for the solventless transcarbonatation of glycerol and ethylene carbonate to render valuable glycerol carbonate and ethylene glycol.**

**Abstract:** Biodiesel is nowadays added in 5–10% $v/v$ to diesel, and its production involves the parallel creation of a vast glycerol amount as a by-product. Despite its many applications, there is a surplus of glycerol (Gly) that has boosted the search for new applications of this compound, now transformed into an industrial synthesis intermediate or platform chemical. Its transcarbonation is a type of reaction that occurs under mild conditions, using weak or moderate basic catalysts, and allows the parallel production of glycols of industrial interest with high selectivity, such as ethylene glycol. In this research, we have studied the activity of a Tunisian clay rich in inorganic carbonates that give it a weak basic character. The raw clay (RC) has been fully characterized by XRD, FTIR, SEM-EDS and nitrogen porosimetry. Subsequently, it has been employed as a catalyst to react glycerol (G) with ethylene carbonate (EC) to obtain glycerol carbonate (GC) and ethylene glycol (EG). The main operating variables and their effects on glycerol conversion and initial reaction rate were analyzed: catalyst concentration (2–6% $w/w$ glycerol), reagent molar ratio (EC:G 1.5:1 to 3:1), and temperature (80–110 °C). Then, an appropriate kinetic model was selected from the results obtained under various reaction conditions, including the total deactivation of order 1 of the catalyst. The kinetic constant activation energy in this reaction using Tunisian smectite was found to be around 183.3 kJ·mol$^{-1}$. In the second phase of the investigation, we explored the reuse of smectite using the kinetic model to appreciate the effect of cycle-to-cycle deactivation. It can be seen that the kinetic constant of the main reaction generally decreases with the number of cycles at low temperature and goes through a maximum at high operating temperature, while the deactivation constant increases with the number of catalytic cycles. The catalyst shows more stability, in general, at higher temperatures.

**Keywords:** glycerol valorization; ethylene glycol; glycerol carbonate; smectite



## 1. Introduction

With the growing perception for the need for sustainability at global scale, alternative fuels and fuel additives for transport has gained more attention. The intensive consumption

of fossil fuels, the depletion of natural resources and the concomitant environmental concerns drive a progressive change to green processes [1]. This fact is also reflected in the gradual increase in the share of energy use based on renewable sources that has been encouraged throughout the world in the early years of the present century, in particular in the European Union [2]. As a consequence, a fast growth in the biodiesel industry has taken place in the last decades, as it has received worldwide attention as a medium-term alternative to diesel fuel obtained from petroleum. Biodiesel mixtures are chemically simple: they are in essence fatty acid methyl or ethyl esters. They have the texture of a yellow liquid and are immiscible in water, have a high boiling point and a low vapor pressure, and are not very flammable (370 to 440 K), while their densities are lower than that of water ($\sim$0.86 g·cm$^{-3}$). Usually, biodiesel is obtained by transesterification of vegetable oils and/or animal fats, with the concomitant production of glycerol, a by-product whose production increase is calling for novel processes and products based on this platform chemical.

A derivative of glycerol, glycerol carbonate (GC) is a chemical of high interest due to its many applications as a polyvalent green solvent [3] or as a reagent to treat biomass through acid pretreatments, e.g., sugarcane bagasse [4] and rice husk [5]. Thanks to its non-toxic properties, this carbonate has been applied in cosmetic products, for example as a component in nail polish remover gel, emulsifier, as plasticizer, and as humectant. Over the past two years, there has been an increase in the number of patents filed where GC was used in hair (shaping, dyeing) or skin (lightening, bleaching) applications [6].

There are several methods and approaches to synthesize GC from glycerol, which will react with different sources of carbonate, such as $CO/O_2$, $CO_2$, organic carbonate, urea, or phosgene [7]. Glycerol carbonylation with phosgene or CO is a synthetic route which has been used for several years for the production of GC, but due to the high toxicity and the caustic aspect of the reagents, this method of synthesis has been abandoned over time [8,9]. Recently, studies have clearly mentioned that the use of $CO_2$ as a source of carbonate for the synthesis of GC is an ideal approach. However, this method suffers from a very limited overall activity due to the difficulty in activating the $CO_2$. Likewise, when using urea as a reagent, the by-product $NH_3$ needs to be removed continuously during the reaction. Lately, various non-toxic, safer, and environmentally friendly synthetic routes have been developed for the production of GC based on organic carbonates. One route of this kind is the transcarbonation of glycerol with dimethyl carbonate (DMC) [10], which acts as an excellent solvent [11]. Another targeted method for the production of glycerol carbonates described in the literature is the glycerol transcarbonation with cyclic carbonates: ethylene carbonate (EC) or propylene carbonate (PC) [12].

This reaction requires a basic catalyst identical or similar to those employed in the biodiesel process, working at very mild temperatures and atmospheric pressure. Sodium hydroxide (NaOH), potassium hydroxide (KOH), potassium carbonate $K_2CO_3$, and triethylamine are conventional catalysts used to produce biodiesel [13]. Although these homogeneous catalysts can produce high GC yield, they require a notable processing effort to purify the carbonate [14]. In addition, their presence can affect the purity of the fuel and cause environmental impacts.

Heterogeneous catalysts are designed to overcome the disadvantages of using a homogeneous one. For instance, they can reduce the amount of wastewater and provide an easy-handling separation. Although these are high-performing catalysts, they require special effort to achieve their full commercial potential. Mixed metal oxides as $MgO/La_2O_3$, $CaO/MgO$, and $MgO/Al_2O_3$ have been widely used as basic catalysts for the production of GC. Their surface area and composition can be adjusted easily. These catalysts have higher catalytic activity than that of a single component [15–18]. Zeolites modified by ion exchange of alkaline cations have also emerged as interesting solid bases [19,20]. They are known to catalyze reactions that require a base site, such as transesterification reactions. The base strength of an alkali ion exchanged zeolite increases with an increasing electropositivity of the exchange cation, and subsequently the product yield. There is considerable interest in the development of catalysts based on clays. With regard to their environmental aspect,

these materials are one of the most user-friendly solid catalysts. Clays (in particular, smectites) are successfully used in many organic reactions [21]. Transcarbonation has been successfully driven with residues, such as fuel ash [22], with regard to esterification, mineral acids, metal exchangers and heteropolyacid-supported montmorillonite [23,24]. To follow glycerol carbonate production, Hammond et al. used HPLC [25]. José et al. applied FTIR analysis to quantify GC when its concentration is around 70 to 75% in a mixture of glycerol and GC [26].

Herein, we report the preparation, characterization, and catalytic activity of Tunisian smectite in the transcarbonation reaction of glycerol and ethylene carbonate to glycerol carbonate and ethylene glycol in solventless conditions. The smectite is fully characterized in terms of BET porosimetry, SEM-EDS, FTIR and DRX, while the reaction is followed qualitatively by $^1$H-NMR and, in a quantitative manner, by ion exclusion HPLC (High-performance liquid chromatography). Data are collected from runs performed at diverse catalyst concentrations, reagent molar ratios and temperatures. Subsequently, a step-by-step proposal, fitting and validation methodology is applied to obtain a kinetic model adequate for a wide experimental range. Finally, the stability of the catalyst is studied in depth in a wide temperature interval for four cycles, applying the aforementioned kinetic model now featuring an added first-order deactivation to deepen into the mechanisms underlying the smectite deactivation.

## 2. Materials and Methods

### 2.1. Materials

Diverse reagents have been employed (all purchased from Sigma-Aldrich (Burlington, MA, USA), if no other supplier was specified): extra pure glycerol (99.88% assay grade) from Fischer Chemical (Zurich, Switzerland); ethylene carbonate (synthetic grade, purity 99%) from Scharlau as reactive species. Raw Tunisian clay was used as a catalyst. To calibrate HPLC analysis, next reagents were used: glycerol carbonate (purity ≥ 99.5%) from Sigma-Aldrich, HPLC grade methanol (test grade, 99.99%) from Scharlau Chemie (Barcelona, Spain) and anhydrous ethylene glycol (99.8% purity) from Sigma-Aldrich. Finally, the citric acid ACS reagent (purity ≥ 99.5%) from Sigma-Aldrich was used as an internal standard for ion exclusion HPLC.

### 2.2. Methods

#### 2.2.1. Purification of Clay

The natural clay powder used in this work was taken from the Hidoudi Mountain of the Gabes region, in the south-east of Tunisia [27–31]. The clay purification process went through diverse stages, starting with crushing in an agate mortar until particles of 1 mm at most were obtained. The fine fraction (<2 μm) was obtained according to the standard purification methods of Van Olphen (1963). One kilogram of natural raw clay was dispersed and thoroughly mixed in 300 mL of distilled water to eliminate the remains of animal fossils and plant debris. After cleaning, the clay was dried in an oven at 100 °C for 2 days and, optionally, sieved to obtain a particle size fraction with a particle diameter lower than 100 microns.

#### 2.2.2. Characterization of Clay Catalyst

The crystalline structure of the raw clay (RC) was determined with an X-ray diffractometer (EMPYREAN) using a CuKa1 radiation source (θ = 1.540598 Å). The sample was scanned in the range of 5–79.984° (2θ), the step size of 0.026°.

To determine the main functional groups, FTIR spectra of the catalysts were recorded in the 4500–500 cm$^{-1}$ wavenumber region using a Perkin-Elmer Spectrum 100 device.

The morphology of these solids were studied using scanning electronic microscopy (SEM) (SEM-JEOL-JSM6301-F) with an Oxford INCA/Energy-350 microanalysis system. This apparatus was also used to investigate the elementary composition of the catalyst sample by energy-dispersive X-ray spectroscopy (EDS) at a 133 eV resolution.

BET surface area ($S_{BET}$), volumes of micro ($V_{Ap}$) and mesopores ($V_{mp}$) were estimated using results from $N_2$ adsorption–desorption isotherms at 77 K, which was measured in a Micromeritics ASAP 2020 using samples outgassed at 613 K to a vacuum of $<10^{-4}$ to ensure a dry clean surface, free of loosely adsorbed species. The $N_2$ cross-section area of reference was 0.16 $nm^2$. To perform micro- and mesoporosity analysis, the BJH method on the $N_2$ isotherm was applied [32].

### 2.2.3. Transcarbonation Reaction

The transesterification of glycerol with ethylene carbonate was carried out in a glass reactor at atmospheric pressure, ensuring the presence of only one phase reaction by working at temperatures higher than 75 °C. Typically, glycerol (50 mmol) and an adequate amount of ethylene carbonate of 75 to 150 mmol were introduced into a 50 mL round bottom flask submerged in a glycerol bath, placed on and IKA Yellow Line TC3 plate provided with PID temperature control and stirring control. The reaction mixture was brought to a temperature of between 80 °C and 110 °C under magnetic stirring at 400 r.p.m. until a clear homogeneous phase was obtained. When the working temperature was reached, a certain mass of raw Tunisian clay (from 2% to 6% of the limiting reactor mass, depending on the system studied) was added. Stirring was performed with a disc-shaped magnet to avoid attrition of the catalyst by reducing the contact surface between the magnet and the round-bottom flask. The reactions of interest are shown in Figure 1.

**Figure 1.** Reaction of smectite-catalyzed of glycerol and ethylene carbonate transcarbonation to ethylene glycol and glycerol carbonate.

The kinetic runs were performed for four hours, withdrawing a total of 14 samples (100 μL each) during this period. To follow the evolution of the chemical composition of the reacting liquid, an ion exclusion HPLC method described in a part of Section 2.2.5 was applied. Interestingly, the analytic procedure allowed for the quantification of all relevant chemical compounds, as peaks for all of them were obtained with high to sufficient resolution.

### 2.2.4. Catalyst Reusability Procedure

For the catalyst reuse test, the tested catalyst was removed after finishing the catalyst reaction using a centrifuge, and washed several times (20 min per washing step, 400 rpm magnetic agitation, 40 °C): 2 times with acetone and 2 times with methanol. After each washing step, solid–liquid separation was carried out with a centrifugation at 9000 g for 15 min at a temperature of 20 °C. The washed catalyst was immediately employed for another catalytic run directly without further processing. This procedure was repeated 4 times to study the reusability of our catalyst.

### 2.2.5. Analytical Techniques
Qualitative Analysis

A qualitative assessment of the compounds present in the reaction liquid was analyzed using nuclear magnetic resonance (NMR) BRUKER Ultra Shield Plus 400 MHz. In this case, the sample for the $^1$H-NMR analysis was prepared by mixing 50 μL of liquid sample with 1000 μL of deuterated DMSO-d6 in an 1.5 mL microtube, centrifuging it at 14,000 r.p.m.

(20,913× $g$ considering the rotor) and filtering through a 0.22 μm 13 mm disc. Finally, 750 μL of the filtrate are placed in a NMR tube before sending it to a central NMR facility.

Quantitative Analysis

For ion exclusion HPLC, the initial 250 μL sample was subjected to centrifugation at 14,000 r.p.m. to remove most solids. Afterwards, 100 μL were taken and diluted 25-fold with an aqueous solution containing 8 g/L of citric acid (internal standard). The diluted sample was filtered through a 0.22 μm 13 mm disc and analyzed by ion exclusion HPLC in a JASCO 2000 equipment set with a refractive index detector. Milli-Q acidic water (0.005 N $H_2SO_4$) was the mobile phase selected, flowing at a constant flow rate of 0.5 mL/min. The separation of the peaks of each component was carried out using a Rezex ROA-Organic Acid $H^+$ (8%) column (150 × 7.80 mm). Afterwards, the peak areas were taken and glycerol internal standard curves at several reagent molar ratios were applied, calculating the glycerol conversion with the following equation:

$$X = \frac{C_{0,Gly} - C_{Gly}}{C_{0,Gly}} \qquad (1)$$

where $C_{0,Gly}$ and $C_{Gly}$ are the glycerol concentrations at initial time and at a certain time, respectively.

Once raw glycerol conversions were known, Origin 2021 software was applied for smoothing data by interpolation using adequate hyperbolic functions, thus reducing the experimental random error. This step provides corrected values of glycerol conversion, and the data employed in the kinetic modelling.

### 2.2.6. Statistical Methods

Kinetic data retrieved at diverse catalyst concentrations, reagent molar ratios and temperature values are employed to propose, fit and validate a potential kinetic model based on the assumption of an elemental bimolecular reaction between glycerol and ethylene carbonate. Fitting of the kinetic model to diverse datasets was performed using a combination of a numerical integration of the ODE of the model with a variable step Euler method and a non-linear regression algorithm named NL2SOLV, an evolved form of the Levenberg-Marquardt gradient method. These algorithms were implemented in the Aspen Custom Modeler v 12.1 software.

The kinetic model was first fit to data at 80 °C, ethylene carbonate to glycerol molar ratio (M) equal to 2 and catalyst concentration ($C_{cat}$) from 2 to 6% *w/w* of glycerol (9–27 g·$L^{-1}$) to retrieve a first value for the kinetic constant k and another for the $K_{cat}$ parameter. This parameter value was fixed for the rest of the fitting process, serving as a validating tool. A second fitting stage included datasets at 100 °C at a fixed $C_{cat}$ (2% *w/w*) to have a second value for k and an approximate value of the activation energy $E_a$ using the Arrhenius equation (Equation (2)). Finally, for the kinetic modelling with fresh catalyst, a final fitting and validation stage was the multi-temperature fitting of the model to retrieve the optimal value for the activation energy and the preexponential Arrhenius term. This approach allowed for the obtention of a kinetic model valid for the wide experimental reality studied. For the stability studies, a more complex kinetic model was considered as deactivation was present and could be modelled by an additional first-order kinetic equation, but a model fitting was simpler: the model was fitted to one experiment at a time to collect the kinetic parameters and analyze their evolution with the reuse cycles.

$$k = exp\left(k_0 - \frac{E_a}{R}\frac{1}{T}\right) \qquad (2)$$

where $E_a$ is the activation energy, $R$ is the ideal gas constant (also known as the universal gas constant, or molar gas constant), its exact value is 8.31 J·$K^{-1}$·$mol^{-1}$, $T$ is the temperature in K, $k$ is the rate constant of the reaction, the proportionality constant $k_0$ is the pre-exponential

factor, or the frequency factor that takes into consideration the frequency at which reactive molecules collide and the probability that a collision will lead to a successful collision for reaction.

Apart from the usual physical considerations (positive values for the kinetic constants, positive value for the activation energy within the 10–300 kJ·mol$^{-1}$ interval), several statistical criteria were analyzed to determine if each fitting was correct and, in the end, if the kinetic model was adequate, either in with fresh or with reused catalysts. Statistical analysis was based on the standard error for the kinetic parameters of each model and the goodness-of-fit parameters based on the least squared method. In fact, all goodness-of-fit statistical criteria applied were based on minimizing the sum of variances or squared residuals (RSE) as indicated in Equation (3), where $X_e$ stands for the experimental values of glycerol conversions, while $X_c$ are the conversion values calculated with the kinetic model. A derivate parameter is the standard error of estimate (SEE), a parameter that is calculated by using Equation (4). Fisher's F value (F) is estimated through Equation (5) and should be higher than a threshold value for the given number of data and kinetic parameters (for our case, they are in the range 20–30). A fine fitting means a low value of RSS and, in consequence, for RSE, while the value of Fisher's F should be high and, in any case, higher than the aforementioned threshold values at 95% confidence.

$$RSE = \sum_{i=1}^{N}(X_e - X_c)_i^2 \tag{3}$$

$$SEE = \sqrt{\frac{1}{N}\sum_{i=1}^{N}(X_e - X_c)_i^2} \tag{4}$$

$$F = \frac{\sum_{n=1}^{N}\frac{(X_c)^2}{K}}{\sum_{n=1}^{N}\frac{(X_e-X_c)^2}{N-K}} \tag{5}$$

Finally, to analyze the performance of the model in predicting the evolution of the conversion with processing time, the percentage of explained variable (VE) was the most adequate goodness-of-fit. It can be estimated with Equation (6):

$$VE(\%) = 100 \cdot \left[1 - \frac{\sum_{n=1}^{N} SSQ_i}{\sum_{n=1}^{N} SSQ_{meani}}\right] \tag{6}$$

### 3. Results and Discussion

*3.1. Further Characterization of the Selected Catalyst*

3.1.1. XRD Studies

X-ray diffraction is a non-destructive method used for the qualitative and quantitative analysis of polycrystalline samples. Indeed, it is suitable for identifying the mineralogical composition of clay assemblages. The study of the powder diffractograms of the raw clay RC shows characteristic peaks of the clay minerals of the natural clay as well as the associated minerals (impurities). Raw clay XRD pattern is presented in Figure 2, while relevant data of XRD patterns in collected in Table 1. It displays an intense peak at 3.34 Å corresponding to quartz (Q). The peaks observed at d = 4.25, 3.24, 2.89 and 2.57 Å are due to silica and quartz, major clay components. Aragonite typical intensities at 3.78, 3.51, 2.37 and 2.28 Å are appreciated; they are responsible for its basic behavior. Furthermore, the diffractogram shows peaks at d = 6.37 Å due to kaolinite, which is evidence of the heterogeneous nature of this clay. The presence of two distinct reflections at $d_{001}$ = 10.06 Å (2θ = 8.78), $d_{002}$ = 6.37 Å (2θ = 13.89), $d_{002}$ = 4.25 Å (2θ = 20.87°) indicates that the clay is a smectite. The peaks at 5.02, 2.57 and 3.34 Å suggest illite presence.

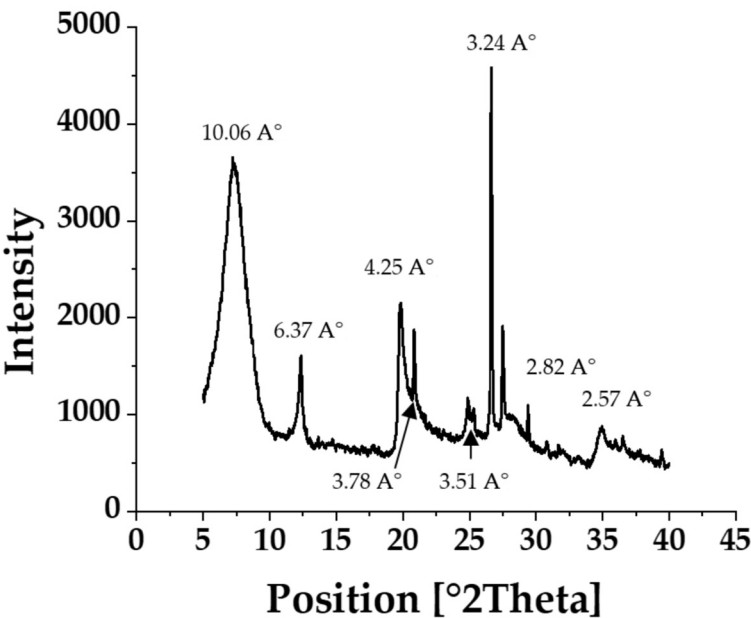

**Figure 2.** XRD patterns of raw clay RC.

**Table 1.** Data of XRD patterns of raw clay RC.

| Pos. [°2Th.] | Height [cts] | FWHM [°2Th.] | d-spacing [Å] | Rel. Int. [%] |
|---|---|---|---|---|
| 8.78 | 666.60 | 0.64 | 10.06 | 13.37 |
| 13.89 | 34.90 | 0.70 | 6.37 | 0.70 |
| 15.81 | 106.83 | 0.11 | 5.60 | 2.14 |
| 17.66 | 325.30 | 0.05 | 5.02 | 6.52 |
| 19.82 | 1084.46 | 0.29 | 4.47 | 21.75 |
| 20.87 | 1258.72 | 0.08 | 4.25 | 25.25 |
| 23.50 | 595.47 | 0.14 | 3.78 | 11.94 |
| 25.36 | 1107.46 | 0.17 | 3.51 | 22.21 |
| 26.66 | 4985.60 | 0.14 | 3.34 | 100.00 |
| 27.52 | 903.13 | 0.17 | 3.24 | 18.11 |
| 30.83 | 497.44 | 0.14 | 2.89 | 9.98 |
| 32 | 559.63 | 0.17 | 2.79 | 11.22 |
| 33.17 | 513.67 | 0.35 | 2.70 | 10.30 |
| 34.81 | 1125.76 | 0.17 | 2.57 | 22.58 |
| 36.57 | 627.99 | 0.17 | 2.45 | 12.60 |
| 37.88 | 405.44 | 0.23 | 2.37 | 8.13 |
| 39.48 | 488.74 | 0.08 | 2.28 | 9.80 |

### 3.1.2. FTIR Analysis

FTIR spectrometry supplements X-ray diffraction. Absorption bands at 985 $cm^{-1}$ and 3704 $cm^{-1}$ are consistent with XRD, indicating the presence of kaolinite in clay. In contrast, the hectorite spectrum shows in addition to the main OH stretching band near 3704 $cm^{-1}$ an absorption near 3617 $cm^{-1}$, assigned to H–O–H stretching vibrations of water molecules weakly hydrogen bonded to the Si–O surface. The band at 1627 $cm^{-1}$ is attributable to the flexion of the H-OH bonds of the structural water molecules, and that at 794 $cm^{-1}$ and 689 $cm^{-1}$ to the bending vibrations of the Al-Mg-OH and Al-Fe-OH groups, indicating the

presence of a smectite. The OH bending region of dioctahedral smectites often provides valuable information on the composition of the octahedral sheets. Two peaks at 985 cm$^{-1}$ (Al$_2$OH) and 917 cm$^{-1}$ (Fe$_2$OH) reflect partial substitution of octahedral Al by Mg the smectite clay. The presence of the 3617 cm$^{-1}$, 917 cm$^{-1}$ and 985 cm$^{-1}$ bands indicates that the smectite is dioctahedral. The band around 794 cm$^{-1}$ is due to the presence of illite. They confirm the presence of smectite, kaolinite and illite in the studied clay, as indicated by the DRX analysis [33]. The relevant spectrum is showed in Figure 3.

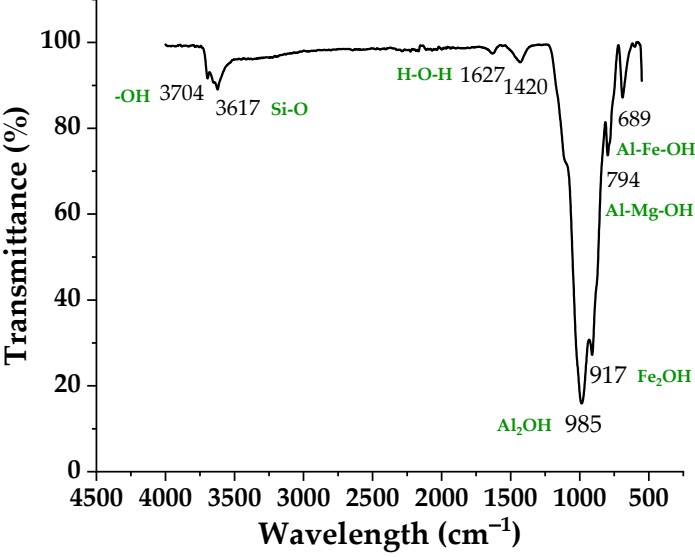

**Figure 3.** FTIR spectra of raw clay RC.

### 3.1.3. Scanning Electron Microscopy (SEM)-Electron Dispersion Spectroscopy (EDS)

Before SEM observation, the samples were dried at 50 °C for 24 h to avoid water interferences. SEM microscopy leads to the observation of clay texture and the characterization of mineralogical assemblages. SEM images at different magnifications are displayed in Figure 4. Clay forms fine aggregates, platelets and rods. There are poorly crystallized kaolinites and illites, as were also appreciated by Fatimah et al. [34]. The images are logical and confirm previous XRD results, showing carbonates and quartz. Calcite is perceived as highly visible aggregates, while small quartz grains are observed. The images show a clay sheet structure. In Figure 4A, (500× magnification), the clay presents different grain sizes. Increasing the magnification to 4000× (Figure 4B), several scattered aggregates of different shapes were observed. Smectite magnification at 20,000× (Figure 4C) shows that the clay forms exploded sheets. EDS results indicate a high atomic percentage of carbon (19.26–60.95%) depending on the scanned zone of the particle(s), with also high percentages for oxygen (34.83–67.44%). Metal elements are present in the smaller percentages: Si (0.23–17.88%), Al (0.30–7.24%), K (0.07–3.99%), Fe (0.49–2.01%), Ca (0.12–1.92%), Mg (0.37–2.04%), and Na (0.57–0.77%).

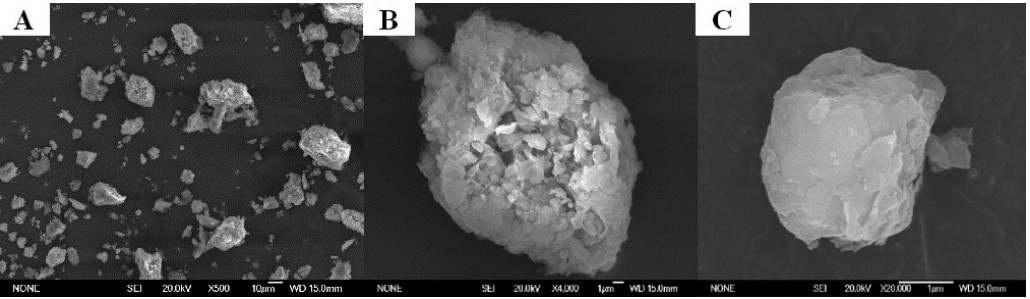

**Figure 4.** SEM micrographs of the smectite at several magnifications of 500× (**A**), 4000× (**B**), 20,000× (**C**).

### 3.1.4. Porosimetry Analysis: BET Specific Surface

The adsorption–desorption isotherms of the raw clay fraction are presented by Figure 5. We note that the nitrogen adsorption–desorption isotherm at 77 K belongs to the type 2 isotherms. It is convex to the y-axis and for which the numbers of moles absorbed seems to tend to infinity when P tends to $P_0$, which is typical when there is a notable influence of macropores. The hysteresis loop is of type $H_3$ indicating that the clay has slit-shaped pores, typical in a lamellar structure. The values of the BET-specific surface of the Tunisian smectite studied, determined from adsorption, is represented in Table 2. This clay has a large specific surface area, approximately 74.235 $m^2 \cdot g^{-1}$, which can be explained by the diverse mineralogy of clays.

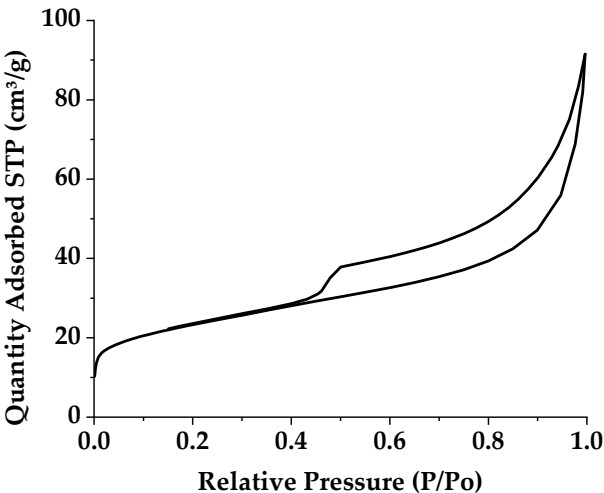

**Figure 5.** Nitrogen adsorption–desorption isotherms of raw smectite.

**Table 2.** Porosimetry characteristics of raw smectite.

| | |
|---|---|
| | BET Surface Area: 74.235 |
| | t-Plot Micropore Area: 11.963 |
| **Surface Area (m²/g)** | BJH Adsorption cumulative surface area of pores between 17.000 Å and 3000.000 Å diameter: 45.847 |
| | BJH Desorption cumulative surface area of pores between 17.000 Å and 3000.000 Å diameter: 71.768 |
| | Single point adsorption total pore volume of pores less than 5369.337 Å diameter at P/Po = 0.996403834: 0.141 |
| | t-Plot micropore volume: 0.008 |
| **Pore Volume (cm³/g)** | BJH Adsorption cumulative volume of pores between 17.000 Å and 3000.000 Å diameter: 0.119 |
| | JH Desorption cumulative volume of pores between 17.000 Å and 3000.000 Å diameter: 0.138 |
| | Adsorption average pore width (4V/A by BET): 76.3032 |
| **Pore Size (Å)** | BJH Adsorption average pore diameter (4V/A): 104.267 |
| | BJH Desorption average pore diameter (4V/A): 76.954 |

### 3.2. Qualitative and Quantitative Reaction Analysis

The validity of the formation of glycerol carbonate is determined by carrying out a $^1H$ NMR spectroscopic analysis, and the peaks obtained have been matched with the characteristic peaks corresponding to glycerol carbonate (Figures 6 and 7). During the transesterification reaction, when Gly begins to react with ethylene carbonate, several peaks appear showing the incomplete transesterification of glycerol; therefore, a reaction

sample containing appreciable concentrations of both reactants and products is shown in Figure 6, in particular the expanded 3.2–5.3 ppm region of the NMR spectrum. The signals in the 4.5–3.2 ppm regions of the spectrum belong to glycerol, whose $H_2$ protons appear around 3.3 ppm and $H_3$ appear around 4.5 ppm, but there is also an overlap with the signals of the glycerol carbonate, which can be observed according to the literature at 4.3 ppm, 3.69 ppm and 3.68 ppm [35–37]. In this spectrum, glycerol carbonate signals were observed at 4.8–3.5 ppm. Thus, the maximum characteristics of [1]H NMR glycerol carbonate are [1]H (DMSO-d6, 500 MHz), δ (ppm), a multiplet around 4.8 attributable to $H_6$ in cyclic CH, a triplet and a quadruplet consecutively around 4.1 and 4.3 attributable to $H_7$ in cyclic $CH_2$, two split doublets around 3.5 and 3.7 ($H_5$ in non-cyclic $CH_2$), and a large singlet around 5.3 attributable to the proton of the hydroxyl group ($H_4$ in OH). Two singlets appear around 4.5 and 3.4, respectively, which can be attributed to the $H_1$ and $H_8$ protons (in $CH_2$, $CH_2$) of ethylene carbonate and ethylene glycol molecules.

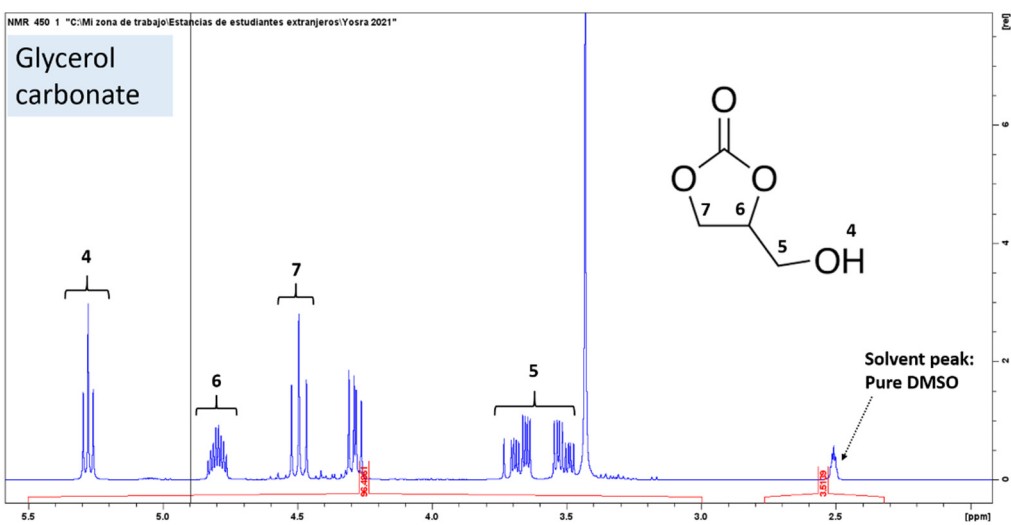

**Figure 6.** Expanded region (3.2–5.3 ppm) of [1]H NMR spectrum of a reaction sample containing the mixture Gly, EC, GC and EG using DMSO-d6 as solvent.

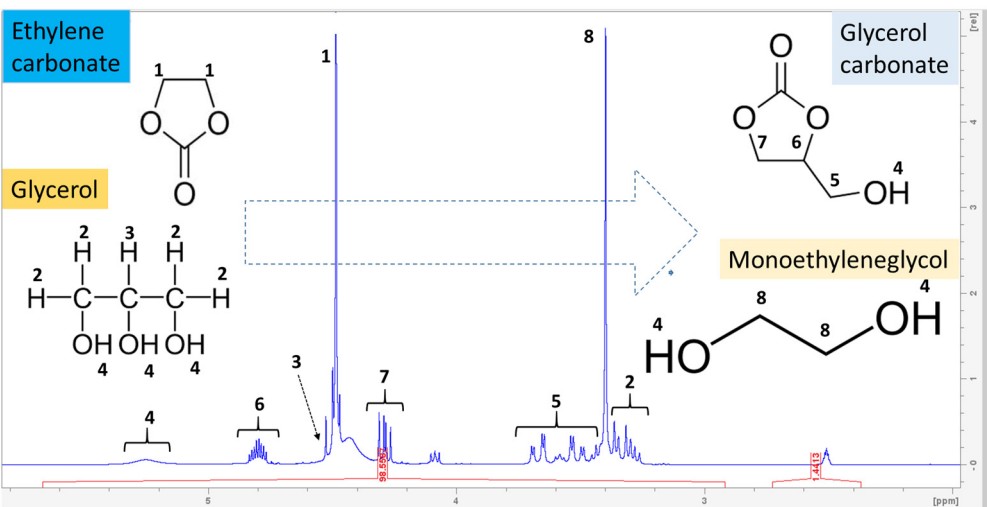

**Figure 7.** Expanded region (3.2–5.3 ppm) corresponding to the [1]H NMR spectrum of pure glycerol carbonate (GC) in DMSO-d6.

Glycerol transcarbonatation with organic carbonates occurs via an intermediate species resulting from the attack of the glycerol oxide anion on the carbonate group carbon atom. This intermediate turns rapidly into glycerol carbonate and monoethylene glycol [38].

In our case, the esterification of glycerol with ethylene carbonate gives two intermediate compounds. The first intermediate $A_1$ is the 2,3-dihydroxypropyl (2-hydroxyethyl) carbonate, as per its structure shown in the chromatogram in Figure 8. Moreover, GC further reacts with EC in excess, generating compound $A_2$ (2-oxo-1,3-dioxolan-4-yl)methyl (2-hydroxyethyl) carbonate, its structure is represented Figure 8.

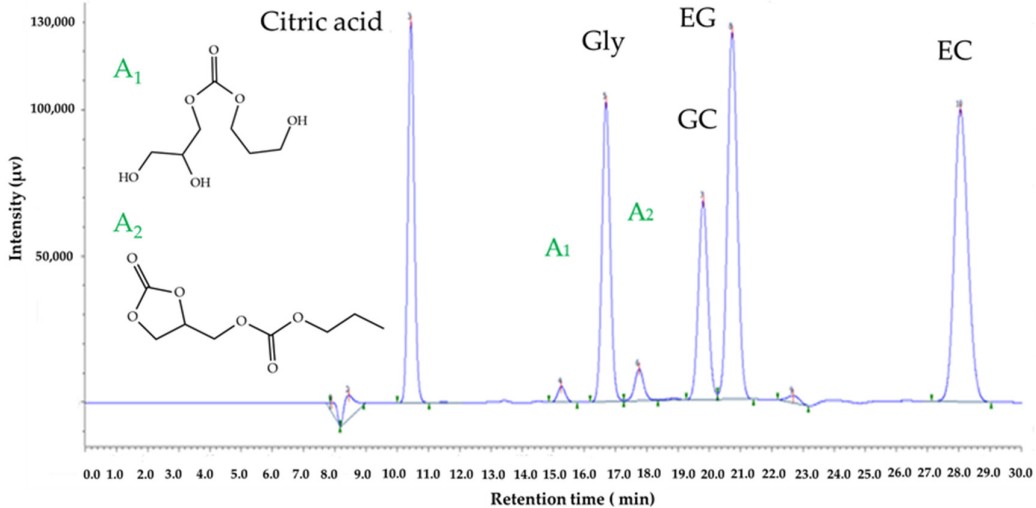

**Figure 8.** Chromatogram of the reaction between EC and Gly including all species, the internal standard (citric acid) and minor by-products **$A_1$** and **$A_2$**.

This chromatogram shows a sample ion exclusion chromatogram of a sample taken after 80 min from the reaction medium, produced with an EC/Gly molar ratio of 2 and at 90 °C. Here, the peaks corresponding to each species are labelled. In addition to the reactants EC and Gly and products GC and EG, which are properly identified, it also includes the peak of the internal standard and the reaction intermediate $A_1$, and the further product of reaction $A_2$. Moreover, since no standards were available for $A_1$ and $A_2$ species, the two peaks of minor species appearing at retention times of 15.25 min and 17.74 min, respectively, could be identified following the evolution of their signals with reaction time, as displayed in Figure 8. Thus, the figure shows that the analytical method used here makes it possible to achieve a very high peak resolution, monitoring all relevant species both qualitatively and quantitatively.

### 3.3. Catalytic Activity: Transesterification of Glycerol with Ethylene Carbonate

The activity test was performed to obtain carbonated glycerol in good yield. The common conditions used were by changing the EC/Gly molar ratio and the catalyst concentration. To ensure miscibility between Gly and EC, 80 °C was the lowest temperature chosen for this reaction system, avoiding the formation of two liquid phases, while 110 °C was chosen as the highest temperature to avoid a reaction driven mainly by heat. Therefore, the higher reaction rate and steeper compositional changes are due to the elevation of the reaction temperature, as it is shown in Figure 9 below, for runs performed with a concentration of catalyst of 2% *w/w* glycerol and a molar ratio EG/Gly equal to 2. At 110 °C the reaction finishes at 4 h, with an almost quantitative conversion of glycerol and a very high yield to glycerol carbonate and ethylene glycol (almost 100% in both cases). Similar results have been achieved in other conditions [39,40].

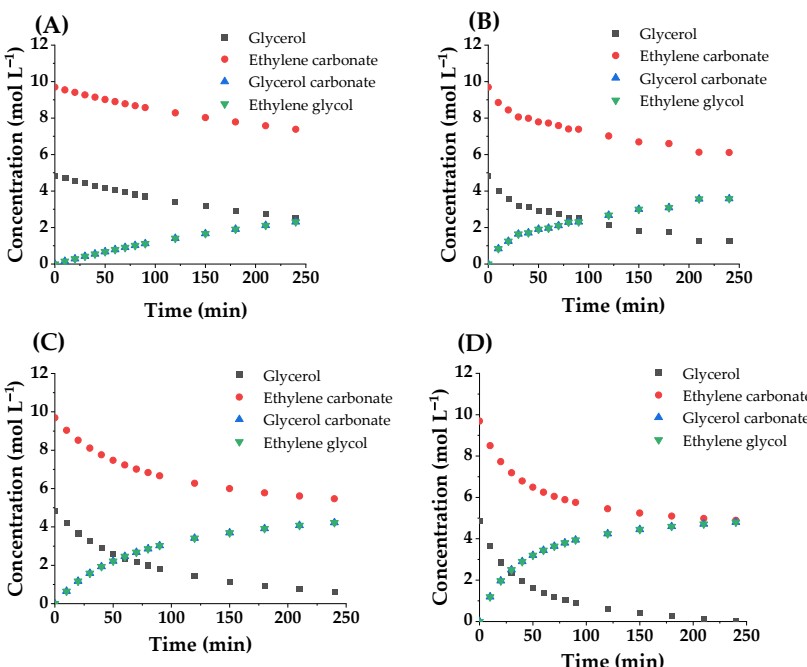

**Figure 9.** Variation of concentration of reagents and products with time at tested temperatures: (**A**) 80 °C; (**B**) 90 °C; (**C**) 100 °C and (**D**) 110 °C.

### 3.3.1. Effect of the Catalyst Concentration

The effect of the dosage of the catalyst on the yield of glycerol carbonate was studied by varying the concentration of RC, by selecting values as 2%, 4%, and 6% at constant EC/Gly molar ratio of 2/1 and reaction temperature of 80 °C; being the reaction runs performed for 4 h. The GC yields and initial overall reaction rate are displayed in Figure 10. As shown in Figure 10A, the glycerol conversion and GC yield increased constantly with the increase of the catalyst amount in the range from 2% to 6% *w/w*. The glycerol conversion was zero at 4 h when no catalyst was added to the reaction mixture (negative control); whereas it reached 47% with 2% *w/w* and 52% with 6% of the catalyst. These results indicate that the catalyst amount of 2.0% was enough for the transesterification of glycerol with ethylene carbonate. In theory, a higher-concentration catalyst will contain a greater number of active sites, so it should lead to a higher conversion of glycerol.

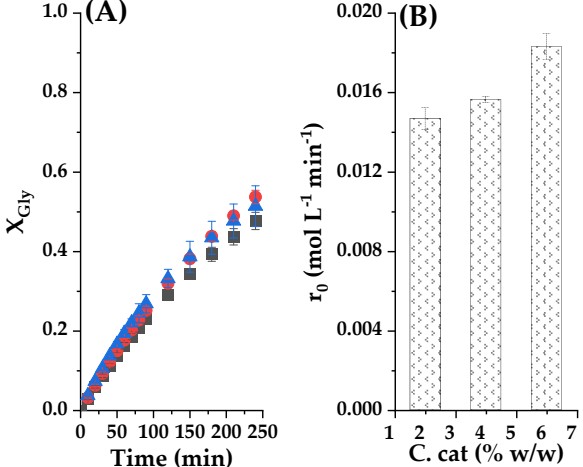

**Figure 10.** (**A**) Catalyst load effect on the conversion of Gly as a function of reaction time, where ■ represents 2% catalyst loading; ● 4% catalyst loading and ▲ a 6% catalyst loading. (**B**) Initial observed reaction rates for these experiments. Common experimental conditions: 80 °C and CE/Gly molar ratio equal to 2.

However, the distribution of the reagents on the catalyst surface depends also on the amount of solid available, while catalysis in this case are a surface phenomenon where saturation phenomena can take place, thus limiting the maximal reaction rate [10,14]. Figure 10B shows the variation of initial reaction rate with catalyst concentration to illustrate this point. It is observed that a high quantity 6% of catalyst gives more active sites for the substrate binding; it slightly increases the rate of the reaction. However, we notice that there is only a minor improvement in initial reaction rate by increasing the quantities of the catalyst beyond 2%, instead of a linear rise of $r_0$ with catalyst concentration which should happen. This observation led us to choose this value as the percentage of clay for subsequent runs.

### 3.3.2. Effect of Molar Ratio

As transcarbonation is a reversible process, it is usual to add an excess of the organic carbonate to ensure a high conversion of glycerol and, as a consequence, a high yield to glycerol carbonate and ethylene glycol, also increasing reaction rates in this solventless system [41,42]. After studying the optimum conditions of load catalyst useful for the reactivity of ethylene carbonate with glycerol, we studied the effect of the molar ratio of these reagents effect at 100 °C, to also have a first evaluation of the temperature effect by comparing to the previous results.

In this case, we evaluated three molar ratios: 1.5:1, 2:1 and 3:1 (ethylene carbonate/glycerol) to see the effect of in the experimental conditions previously established for the catalyst concentration, −2% of raw clay (0.36 g) at 100 °C. In the case of using a molar ratio of 3:1, the conversion of glycerol attained its maximum value of 94% at 4 h, which can be appreciated in Figure 11A. Therefore, the excess of the carbonate reagent is of importance to ensure a high conversion of glycerol and a relatively high average reaction rate during the 4 h reaction. Figure 11B shows that the initial reaction rate is influenced by the molar ratio of the reactants; it can be seen that the fastest reaction will take place when the EC/Gly molar ratio is equal to 1.5:1. Therefore, a larger excess of ethylene carbonate means an excessive dilution of glycerol at zero time that reflects on the reaction rate.

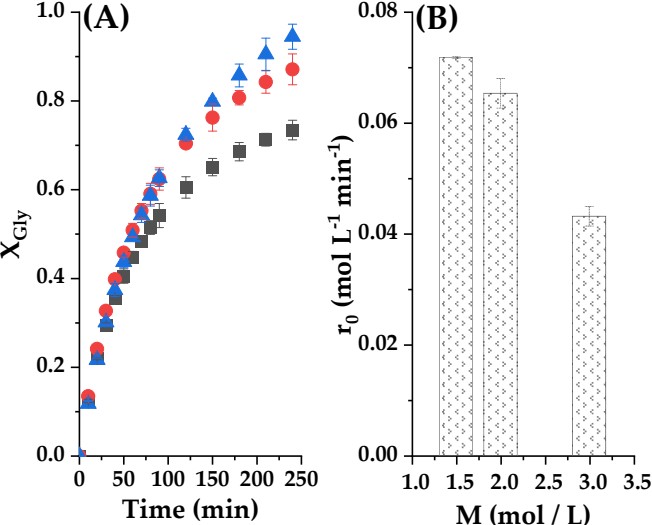

**Figure 11.** (**A**) Initial reagent molar ratio influence setting the reaction temperature at 80 °C and the catalyst mass load at 2%. Here the glycerol conversion versus reaction time with CE/Gly ratio (M) is observed ■ M = 1.5; ● M = 2; ▲ M = 3 (**B**) Initial reaction rate at diverse initial CE/Gly molar ratio.

### 3.3.3. Effect of Temperature

The effect of reaction temperature in glycerol conversion and in the initial rate of transesterification reaction with Tunisian smectite in the range 80–110 °C is presented in Figure 12. As indicated by the collision theory, higher temperatures lead to increased collisions between the reactant molecules, resulting in a higher frequency of reactions. Therefore,

the rising temperature in the interval between 80 °C and 110 °C increased the collision number between reactive molecules, thus improving conversion percentage [11,18,43]. We appreciated the conversion of glycerol at 4 h was about 47% at 80 °C, a conversion that climbed up to 99.3% with at 110 °C, in both cases with 2% *w/w* catalyst/glycerol and EC/Gly molar ratio set at 2. Thus, to ensure a high conversion of glycerol at 4 h, 110 °C was the necessary temperature. In this condition, we achieved this, as well, a very high yield of GC. Figure 12B shows that there is an exponential increase of the initial reaction rate with the increase in the temperature between 80 °C and 110 °C, as can be expected if the Arrhenius equation is valid.

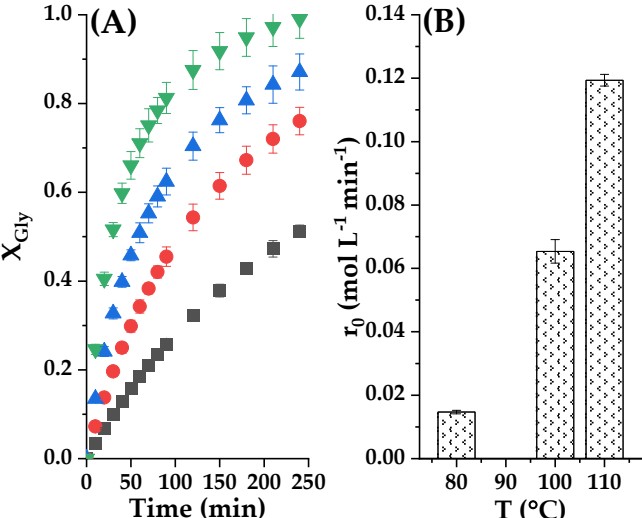

**Figure 12.** (**A**) Temperature influence on the transcarbonatation of Gly with CE at 2% catalyst loading and CE/Gly molar ratio of 2. In this subfigure ■ is for 80 °C; ● 90 °C; ▲ 100 °C and ▼ 110 °C. (**B**) Initial reaction rates at studied temperatures.

In Figure 13, an exponential increase in the initial reaction rate with temperature is evident, an expected Arrhenius tendency. Therefore, after the usual linearization, we can appreciate a linear dependence of $\ln(r_0)$ with the inverse of the absolute temperature $1/T$. As usual, the slope allows for the estimation of an approximate activation energy, obtaining a value of 75.42 kJ·mol$^{-1}$. Values between 32 and 42 kJ·mol$^{-1}$ can be found for the reaction under study when using a homogeneous catalyst, $K_2CO_3$ [44]. If zinc stearate was employed as an interfacial catalyst in similar conditions, an activation energy of 69.2 kJ·mol$^{-1}$ was calculated [45].

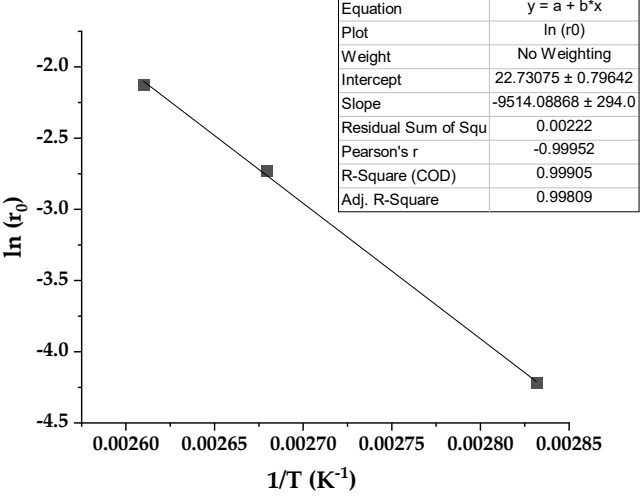

**Figure 13.** Initial reaction rate dependence with temperature (Arrhenius plot).

### 3.4. Catalyst Reusability Tests

Besides the activity, the reuse of a heterogeneous catalyst is indeed very important for all the reactions, including the transesterification in our case. The transcarbonation reaction of glycerol with ethylene carbonate was carried out for 4 reaction cycles under optimal reaction conditions (that is to say, amount of catalyst loaded into the reactor: 2% *w/w* glycerol, EC:Gly (molar ratio: 2:1), temperature 80 °C, 90 °C, 100 °C and 110 °C, the reaction time fixed at 4 h). Having a cycle count of 4 is a well-accepted practice in catalyst stability reports in the open literature [46,47]. Additionally, at the end of each cycle, the catalyst was removed by centrifugation and oven-dried at 35 °C overnight, then reused for the next reaction cycle using fresh glycerol and glycerol carbonate. According to Figure 14, the recycled crude smectite catalyst could be successfully recycled with a slight loss of catalytic activity observed after each reaction cycle. In more detail, the conversion of glycerol decreased from 65.5% to 35.2 at 80 °C, at 90 °C the conversion is observed from 82.1% to 62%, from 85.2% to 65.8% at 100 °C after 4 reaction cycles, and at 110 °C a conversion equal to 98.3% in the first cycle was decreased to 81% in the fourth cycle.

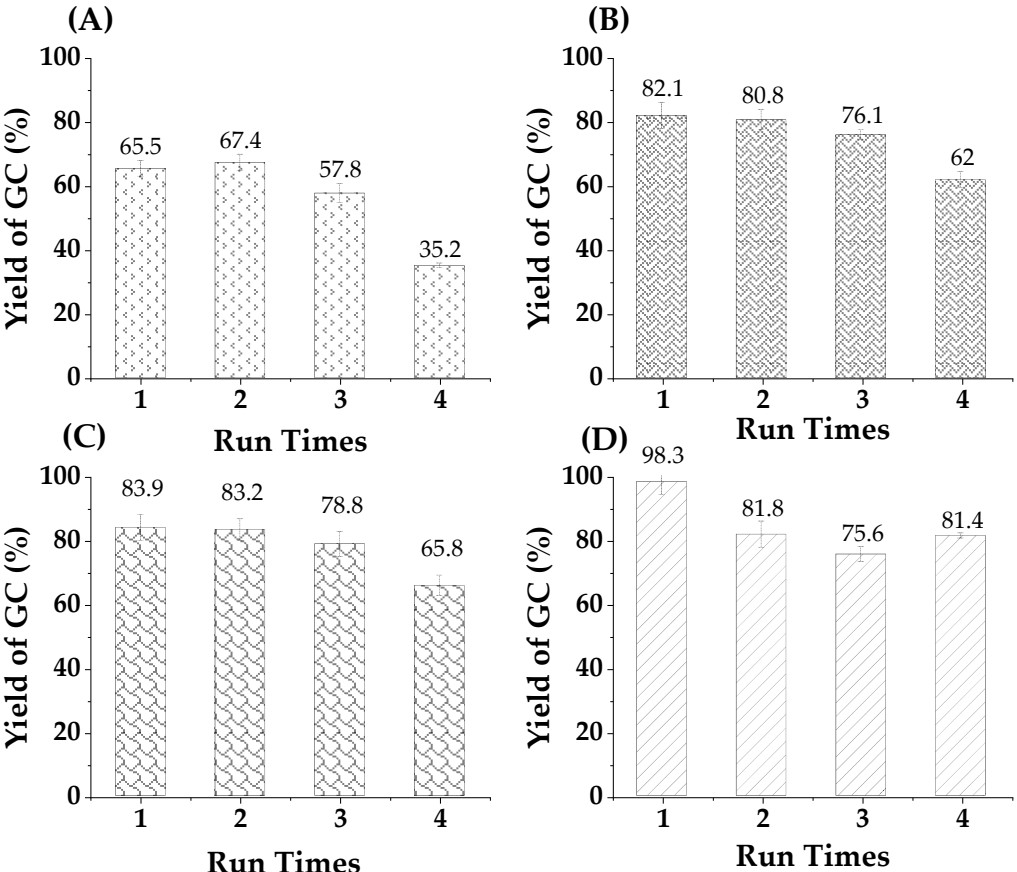

**Figure 14.** Reusability cycles for crude smectite catalyst at different temperatures, (**A**) at 80 °C, (**B**) at 90 °C, (**C**) at 100 °C and (**D**) at 110 °C. The standard % error range is ±0.5 to 5%.

These data reveal that the catalyst can be used for at least four operating cycles with a drop in yield of 10.1% in glycerol carbonate compared to that using the fresh catalyst. However, the trend observed in the yield at 4 h changes notably from the lowest temperature, with a clear grace-period behavior, to the highest temperature, where a stable 80% of the yield with fresh catalyst at 4 h is appreciated in the fourth cycle. This smectite is rich in metallic carbonates, which probably act as a mild basic catalyst. Carbonates are known to reduce their activity in the presence of water and carbon dioxide, due to a progressive neutralization to bicarbonates, with low alkalinity, while water build-up in the reaction zone due to the clay hygroscopicity reduces catalyst activity in the transcarbonatation

reaction by promoting carbonate hydrolysis to $CO_2$ and ethylene glycol [43]. Therefore, operating at temperatures higher than 100 °C ensures a low water environment.

### 3.5. Kinetic Modelling

3.5.1. Model Proposal

The study of kinetics in heterogeneous catalysis is key to the design, scale-up and operation of chemical reactors, and helps in the study of mechanistic details when phenomenological models are proposed and fit to a wide range of experimental data. The first task is to propose the kinetic model itself, based on the observed effects on the initial reaction rate and the conversions and yields at high reaction time values. In this case, we can appreciate a hyperbolic trend of the initial reaction rate with the amount of catalyst, suggesting a reaction taking place in an interface that is saturated with one of the reagents [48,49]. As for these reagents, we assume that the reaction is similar to a esterification or transesterification, so it is an elemental bimolecular reaction and the model is potential in relation with the reagents, being the partial order for the two reagents equal to one, their molecularity or stoichiometric coefficient.

We have found that, in similar reacting systems, we needed to consider the zero order for the carbonate used, since this compound is in large excess compared to glycerol and glycerol carbonate. We appreciated that the carbonate concentration was very high at the liquid–liquid interface, due to mass transfer to the glycerol phase, and that the reaction takes place there, so the order of EC is zero [50,51]. In this case, the proposed model involves a reversible reaction system, but the most notable difference was due to the temperature. When working over 76 °C, EC and glycerol turn to be miscible and the contact between both reagents is ensured. In this case, the concentration of both affects the observed reaction rate in a proportional way, as befits an elemental reaction. As the catalyst is heterogeneous, a model as simple as this one also includes the possibility of no restriction on the reagent adsorption onto the active basic sites, so it is purely the chemical reaction itself as the controlling step. However, the hyperbolic trend of $r_0$ versus catalyst concentration suggest a non-homogeneous distribution of the reagents on the surface of the catalyst. The kinetic model proposed on the premises of our experimental findings is the following:

$$r = \frac{-dC_{glycerol}}{dt} = k \cdot \frac{C_{cat}}{K_{cat} + C_{cat}} \cdot C_{glycerol} \cdot C_{EC} \tag{7}$$

$$\frac{dx}{dt} = k \cdot \frac{C_{cat}}{K_{cat} + C_{cat}} \cdot C_{gly0} \cdot (1 - X) \cdot (M - X) \tag{8}$$

3.5.2. Parameters Estimation

Once the kinetic model was defined, it was fitted to diverse sets of experimental data with the aid of Aspen Custom Modeler v 12.1 software. We used implemented algorithms based on least square estimation for non-linear regression and in the Euler method for ODEs integration, as explained in detail in Section 2.2.6. This methodology was used to obtain optimal values for each parameter, starting the calculations using reasonable initial value given for them by drawing on our experience from modeling other reaction systems [50,51].

The kinetic parameters related to the above model were calculated by nonlinear adjustment coupled with the numerical integration of this model with the experimental data. First, a regression was performed at different concentrations of catalyst ($C_{cat}$) at 80 °C, to determine the kinetic constant k and a parameter $K_{cat}$. Once this last value is retrieved, it was fixed and the same kinetic model was fitted to a second set of experiments at three molar ratios between EC and glycerol from 1.5 to 3. As these runs were performed at 100 °C, the kinetic constant k value increased in an exponential way, as expected. These values are collected in Table 3, together with their errors at 95% confidence. Table 4 compiles all goodness-of-fit parameters, and the goodness of the kinetic model fit to experimental data can also be appreciated in Figure 15. We can appreciate that the model fits perfectly to

data contained in the two datasets: RSE and $S_e$ values are very low while VE% and F are very high.

**Table 3.** Values and standard errors of the kinetic constants estimated after the kinetic fitting to the experimental results obtained at 80 °C and several catalyst concentrations (line 2) and at 100 °C and several reagent molar ratios (line 3).

| Variable | K | $K_{cat}$ |
|---|---|---|
| Catalyst concentration | $4.21 \cdot 10^{-4} \pm 5.45 \cdot 10^{-6}$ | $2.726 \pm 0.222$ |
| Reagent molar ratio | $1.70 \cdot 10^{-3} \pm 2.89 \cdot 10^{-5}$ | 2.726 (fixed) |

**Table 4.** Goodness-of-fit parameters belonging to the fit of the proposed kinetic model to the relevant experimental results: residual sum of squares (RSS), standard error of estimate ($S_e$), percentage of variation explained (VE %) and the Fisher parameter (F).

| Variable | RSS | $S_e$ | VE % | F |
|---|---|---|---|---|
| Catalyst concentration | 0.001631 | 0.62% | 99.85% | 44,206 |
| Reagent molar ratio | 0.03712 | 2.90% | 98.71% | 16,400 |

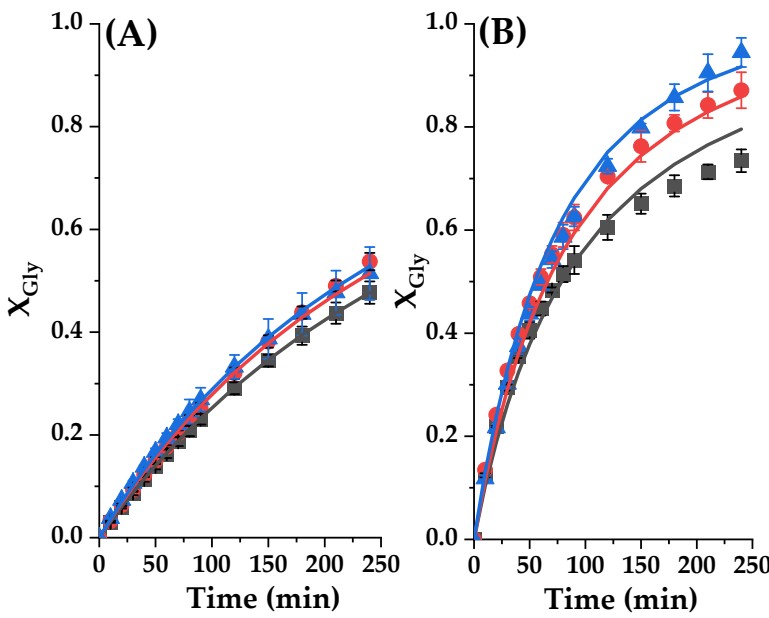

**Figure 15.** Glycerol conversion ($X_{Gly}$) in the transesterification reaction of glycerol with ethylene carbonate catalyzed by Tunisian smectite at diverse experimental conditions. Dots represent experimental data, while lines represent the trend of the model for each experimental data. (**A**) Effect of the catalyst loading at 80 °C with CE/Gly ratio of 2. Gly conversion as a function of reaction time with different catalyst loadings, where ■ represent 2% catalyst loading, ● 4% catalyst loading and ▲ 6% catalyst loading. (**B**) Effect of the initial molar ratio of reactants at 80 °C with catalyst loading 2%. Gly conversion as a function of reaction time with CE/Gly ratio, where ■ represent 1.5 M; ● 2 M; and ▲ for 3 M.

### 3.5.3. Kinetic Model Fitting at Several Temperature Runs

As a further proof of the bounty of the proposed model, a third fitting was performed using datasets from four runs performed at temperatures from 80 to 110 °C, to observe if the kinetic constant is an exponential function of temperature, following the Arrhenius trend-Equation (2). To further validate previous findings, the $K_{cat}$ value was again set at 2.72. Fit results are displayed in Figure 16, while Table 5 compiles key parameters of the kinetic constant together, and the most important goodness-of-fit statistical parameters.

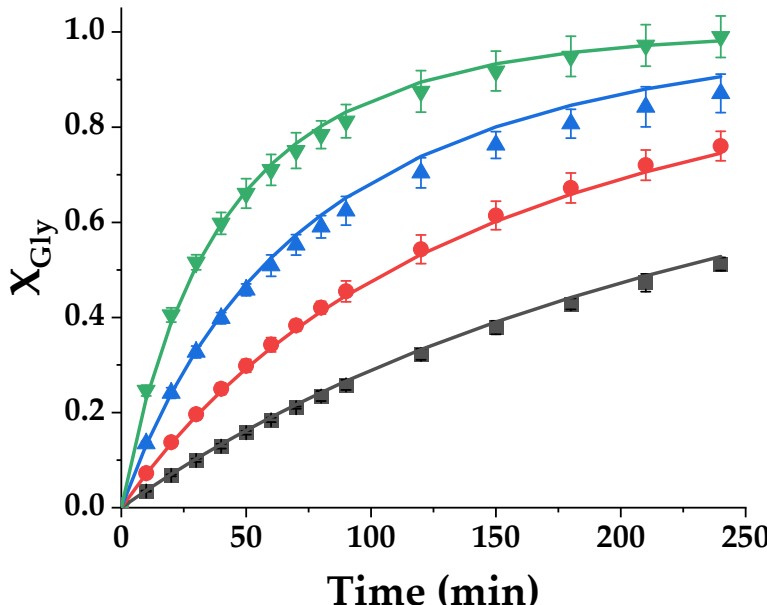

**Figure 16.** Glycerol conversion ($X_{Gly}$) in the transesterification reaction of glycerol with ethylene carbonate catalyzed by Tunisian smectite at different temperatures (■: 80 °C; ●: 90 °C; ▲: 100 °C and ▼: 110 °C).

**Table 5.** Statistical parameters calculated by fitting the sigmoidal model to different sets of experimental data at varying temperatures. Residual sum of squares (RSS), standard error of estimate (Se), percentage of variation explained (VE %) and Fisher's F value (F).

| $k_0$ | $E_a/R$ | RSS | $S_e$ | VE % | F |
|---|---|---|---|---|---|
| $18.09 \pm 0.68$ | $9077 \pm 251$ | 0.0697 | 3.47% | 98.62 | 7444 |

The model fits relevant data at all temperatures in an almost perfect way, and the activation energy optimal value retrieved is 75.42 kJ/mol, a value included in the bibliographic interval of values for this reaction. In consequence, the kinetic model adequate for this chemical reaction under the tested conditions is:

$$\frac{dX_{Gly}}{dt} = exp\left(18.09 - 9077\frac{1}{T}\right) \cdot \frac{C_{cat}}{2.72 + C_{cat}} \times \left(1 - X_{Gly}\right) \cdot \left(M - X_{Gly}\right) \qquad (9)$$

### 3.5.4. Stability and Deactivation

Glycerol and ethylene carbonate are hygroscopic, and this is also evident in the smectite. Therefore, water tends to accumulate in the pores and the inner surface of the clay, and hydrolytic reactions are also possible at these moderate temperatures, competing with the transcarbonation process. Moreover, though all alkaline catalysts are active in this process, their progressive neutralization in the presence of carbon dioxide (in the air) reduces their activity. In this case, carbonate sites become bicarbonate sites, with a notable reduction in their $pK_b$. To further study the deactivation phenomenon, we have applied the proposed kinetic model to datasets at diverse operation temperature for several cycles. In this case, the equation also include a first-order total deactivation of the catalyst [44,49]. Here, $k_d$ is the first-order deactivation kinetic constant. Therefore, the kinetic Equation (8) is modified to include k decreasing in the following form:

$$\frac{dx}{dt} = k \cdot \frac{C_{cat} \cdot exp(-k_d \cdot t)}{K_{cat} + C_{cat}} \cdot C_{gly0} \cdot (1 - X) \cdot (M - X) \qquad (10)$$

Figure 17 shows the variation of the kinetic constant k and the deactivation constant $k_d$ with the number of cycles at different temperatures from 80 to 110 °C. We can observe

that there is an activation from the first to the second cycle at all temperatures, but this is more evident at the highest temperature (a notable increase in the k value). The constant k decreases in the third and fourth cycles until it reaches its original value at all operation temperatures. For example, at 110 °C, which is chosen as the best temperature for this system, the rate constant k increases from $3.7 \cdot 10^{-3}$ $min^{-1}$ after the first cycle to $1.1 \cdot 10^{-2}$ $min^{-1}$ with the end of the second cycle, however it decreases to $3.8 \cdot 10^{-3}$ $min^{-1}$ after the fourth cycle. If we consider the deactivation constant $k_d$, it shows a significant decrease from $4.1 \cdot 10^{-2}$ $min^{-1}$ in the second cycle to $1.6 \cdot 10^{-2}$ $min^{-1}$ in the fourth cycle at 100 °C, while it increases at all other operation temperatures from one cycle to the next one. Again, as expected, the catalyst activation and deactivation is much more evident at the highest temperature, but it is interesting to observe that the deactivation rapidly decreases from one cycle to the other, suggesting a progressive stabilization of the remaining activity of the smectite. This is in agreement with the stabilization of the yield to glycerol carbonate and ethylene glycol at 4 h at 75–80%.

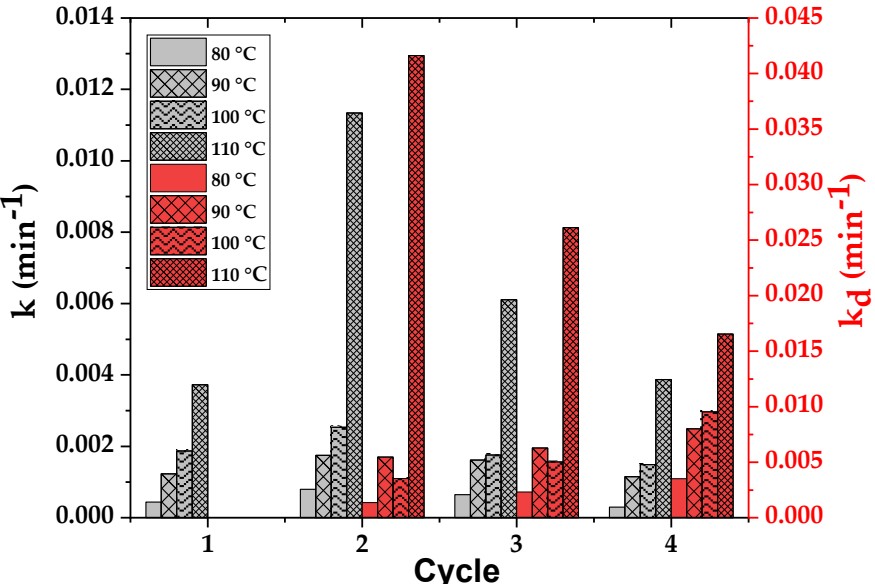

**Figure 17.** Variation of the kinetic constant k and the deactivation constant $k_d$ with the number of cycles at different temperatures from 80 °C to 110 °C.

## 4. Conclusions

The synthesis of glycerol carbonate via the transcarbonation of glycerol with ethylene carbonate using a Tunisian smectite as a catalyst has been studied. The mineralogical and microstructural characteristics of Tunisian raw clay was studied in-depth, showing the abundance of carbonate in its composition. This clay is mainly composed of smectite and, to a lesser extent, of illite and calcite. This catalyst was found to have virtually complete selectivity to transcarbonation products, namely glycerol carbonate and ethylene glycol. A series of kinetic tests were carried out under mild conditions, and a potential second-order kinetic model was proposed and successfully fitted to the experimental data at all operational conditions studied. This same model, adding a first-order total deactivation equation, was also a valid fit to data at several temperatures and for up to four cycles. We found that 110 °C as a fine condition to avoid or reduce inactivation effects on the catalyst, probably due to the absence of water.

**Author Contributions:** Conceptualization, N.B., J.M.B. and M.L.; Data curation, Y.S., M.A.S. and M.L.; Formal analysis, Y.S., I.A.E. and M.L.; Funding acquisition, J.M.B. and M.L.; Investigation, Y.S. and M.A.S.; Methodology, Y.S., I.A.E., M.A.S. and M.L.; Project administration, J.M.B. and M.L.; Software, I.A.E. and M.L.; Supervision, M.L.; Writing—original draft, Y.S. and M.L.; Writing—review and editing, N.B., J.M.B. and M.L. All authors have read and agreed to the published version of the manuscript.

**Funding:** This research was funded by the Spanish Research Agency through the project VALOPACK (PID2020-114365RB-C21) and by the Regional Government of Madrid (2018-T1/BIO-10200). This funding is very gratefully acknowledged.

**Institutional Review Board Statement:** Due to the nature of the research here reported, there was no need for this statement.

**Informed Consent Statement:** As this research article describes a study not involving humans, there is no need for this statement.

**Data Availability Statement:** All relevant data are provided in the figures and tables of this manuscript. The authors will provide any interested reader with tables containing data on figures. Reacting samples in all the conditions reported are available from the authors.

**Conflicts of Interest:** The authors declare no conflict of interest.

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
