# Peer review of "Glycerol Carbonate Solventless Synthesis Using Ethylene Carbonate, Glycerol and a Tunisian Smectite Clay: Activity, Stability and Kinetic Studies"

_applsci, doi:10.3390/app13127182_

Round 1
Reviewer 1 Report
The article needs minor corrections. For example, the equation (2) kinetic model should be explained with each term; the equation (4) SEE is not right, 1/N should be inside of square root; Each letters in the equations should be clarified...
Need minor corrections.
Author Response
We are grateful to the reviewer for his/her careful revision and comments. They have aided us to improve the original manuscript. Please, see our responses (in blue) following each comment (in black); the texts we added in the revised manuscript are contained in this document in green color.
The article needs minor corrections. For example, the equation (2) kinetic model should be explained with each term; Thank you for the suggestion; we have explained each term now. The equation (4) SEE is not right, We have modified SSE by SEE 1/N should be inside of square root; (Corrected, thank you for the observation) each letters in the equations should be clarified... In section 2.1, a more detailed explanation has been included for each abbreviation contained in the equations, thank you for the observation. We add this paragraph to the manuscript
where Ea is the activation energy, R is the ideal gas constant (also known as the universal gas constant, or molar gas constant), its exact value is 8.31 J⋅K−1⋅mol−1, T is the temperature in K, k is the rate constant of the reaction, The proportionality constant k0 is the pre-exponential factor, or the frequency factor that takes into consideration the frequency at which reactive molecules collide and the probability that a collision will lead to a successful collision for reaction.

Reviewer 2 Report
This paper describes the synthesis of glycerol carbonate using Tunisian smectite clay. Several parameters were investigated. The paper is well-written and well-organized. Some minor issues need to be corrected:
1) The abbreviation HPLC should be given in its first appearance in the text.
2) Section 2.2.1.: "Purification of clay the natural clay powder used in this work was taken from the Hidoudi Mountain of Gabes region......." Delete "Purification of clay".
3) Section 2.2.1.: "...... the support clay was ......" Delete "support".
4) Delete comma in 14,000 rpm.
5) Figure 7: The structure of A1 is wrong. There is one additional carbon.
6) Figure 7: Please check the structure of A2. It seemed wrong to me. Besides, reference 13 does not mention this compound. Please revise you discussions on this molecule.
7) Figure 6/8: The NMR spectra can be presented in a better way. Make it more legible (axes). It is better to mark hydrogens with letters or numbers than to circle them (Figure 6).
8) Page 17: ".....conversion equal to 98.3% in the first cycle was
increased to 81% in the fourth cycle." I think "increased" should be changed to "decreased".
9) Did you purify glycerol carbonate? How did you remove ethylene glycol and unreacted monomers?
The level of English is good but there are some typos. Some are noted above but it is good to check the whole paper.
Author Response
Response to Reviewer 2
We are grateful to the reviewer for his/her careful revision and comments. They have aided us to improve the original manuscript. Please, see our responses (in blue) following each comment (in black); the texts we added in the revised manuscript are contained in this document in green color.
Accept after minor revision (corrections to minor methodological errors and text editing)
This paper describes the synthesis of glycerol carbonate using Tunisian smectite clay. Several parameters were investigated. The paper is well-written and well-organized. Some minor issues need to be corrected:
- The abbreviation HPLC should be given in its first appearance in the text. ( Thank you for the comment, we have put the abbreviation of HPLC in the first appearance in the text)
- Section 2.2.1.: "Purification of clay the natural clay powder used in this work was taken from the Hidoudi Mountain of Gabes region......." Delete "Purification of clay".( We have deleted, thank you for the observation)
- Section 2.2.1.: "...... the support clay was ......" Delete "support". (Deleted)
- Delete comma in 14,000 rpm. (Deleted)
- Figure 7: The structure of A1 is wrong. There is one additional carbon. It has been corrected, thank you. As this figure has been eliminated, the structure of the intermediate is corrected (the additional carbon has been eliminated) and their structures are represented in the new Figure 8 that contains the chromatogram of a reaction sample.
- Figure 7: Please check the structure of A2. It seemed wrong to me. Besides, reference 13 does not mention this compound. Please revise you discussions on this molecule.
Figure 7 has been eliminated, as the mechanism is already presented in the literature by Jesus Esteban et al [53]. The mechanism is described by a very detailed paragraph which mentions the two intermediate compounds produced following an excess of organic carbonate and which are A1 is the 2,3-dihydroxypropyl (2-hydroxyethyl) carbonate and the compound A2 is the (2-oxo-1,3-dioxolan-4-yl)methyl (2-hydroxyethyl) carbonate. Their structures are shown in Figure 8 which corresponds to a sample chromatogram of the reaction between EC and Gly and which is including this minor by-products A1 and A2.
- Figure 6/8: The NMR spectra can be presented in a better way. Make it more legible (axes). It is better to mark hydrogens with letters or numbers than to circle them (Figure 6).
Thank you for your comment, we tried to improve the spectrum and put the protons with numbers to clarify it more (it is presented in the manuscript)
- Page 17: "...conversion equal to 98.3% in the first cycle was increased to 81% in the fourth cycle." I think "increased" should be changed to "decreased".
We have corrected it. Thank you for the observation.
- Did you purify glycerol carbonate? How did you remove ethylene glycol and unreacted monomers?
Thank you for the suggestion. We did not purify the glycerol carbonate because the main aim of this paper is to study the catalytic capacity of the smectite, on one side, and the reaction kinetics under its action on the other. In any case, we are interested, and working on the purification of the product in the presence of an excess ethylene glycol. This is feasible under mild vacuum conditions (10-20 mmHg) as the boiling points of glycerol carbonate and ethylene carbonate, in excess, are 354 °C and 243 °C respectively. At that same pressure (1 atm) glycerol and ethylene glycerol boiling points are 294 °C and 197 °C. Therefore, the absence of glycerol due to a very high or quantitative conversion ensures a relatively simple separation of the volatiles (ethylene glycol and ethylene carbonate) from the target product, which has a much higher boiling point

Reviewer 3 Report
1. In the Materials and Methods section, 2.2.4 should be supplemented with specific test methods, some key parameters are missing, e.g., centrifugation rate, number of washes, detection of recoveries, etc.
2. In Figure 2, adding the PDF card information of the standard substances as a control analysis is suggested.
3. In Figure 3, the scale and Figure 8 are ambiguous.
4. in 3.4, "...at 110 °C a conversion equal to 98.3% in the first cycle was increased to 81% in the fourth cycle." Why was the increase to 81%.
5. In this study, the qualitative detection of the product glycerol carbonate is essential, the authors used a mixture to do NMR analysis is not very accurate, it is recommended that glycerol carbonate is separated out for separate NMR analysis or mass spectrometry is better.
6. The citation of [37] does not appear in the article. " ...of which they are observed according to the literature at 4.3 ppm, 3.69 ppm and 3.68 ppm, [13-35]." Many irrelevant literature is cited here and the authors are advised to double check.
Format and syntax can be further improved.
Author Response
Response to Reviewer 3
We are grateful to the reviewer for his/her careful revision and comments. They have aided us to improve the original manuscript. Please, see our responses (in blue) following each comment (in black); the texts we added in the revised manuscript are contained in this document in green color.
- Reconsider after major revision (control missing in some experiments)
- In the Materials and Methods section, 2.2.4 should be supplemented with specific test methods, some key parameters are missing, e.g., centrifugation rate, number of washes, detection of recoveries, etc.
We are very grateful for the suggestion. We have added some key parameters and changed the previous paragraph to this paragraph in section 2.2.4. Catalyst reuse procedure
“For the catalyst reuse test, the tested catalyst was removed after finishing the catalyst reaction using a centrifuge, washed several times: 2 times with acetone and 2 times with methanol each time our catalyst is washed, the separation is carried out with a centrifugation of 9000 g of force for 15 min at a temperature of 40°C overnight, It then undergoes another catalytic reaction directly without further processing, this procedure was repeated 4 times to study the reusability of our catalyst.”
- In Figure 2, adding the PDF card information of the standard substances as a control analysis is suggested.
Thank you for the suggestion. Regrettably, as the analysis have been performed in a central facility, we do not access to the PDF card information.
- In Figure 3, the scale and Figure 8 are ambiguous. (We have done it in the manuscript)
In figure 3 we made some changes in the corresponding text, we tried to add some paragraphs to better explain the FTIR analysis of our smectite.
“FTIR spectrometry has been used to supplement other studies as it is a complementary technique to X-ray diffraction. Absorption bands at 985 cm-1 and 3704 cm-1 are consistent with XRD, indicating the presence of kaolinite in clay In contrast, the spectrum of hectorite shows in addition to the main OH stretching band near 3704 cm-1 an absorption near 3617 cm-1, assigned to H–O–H stretching vibrations of water molecules weakly hydrogen bonded to the Si–O surface. The band at 1627 cm-1 is attributable to the flexion of the H-OH bonds of the structural water molecules and that at 794 cm-1 and 689 cm-1 to the bending vibrations of the Al-Mg-OH and Al-Fe-OH groups, indicating the presence of a smectite. The OH bending region of dioctahedral smectites often provides valuable information on the composition of the octahedral sheets. Two peaks at 985 cm-1 (Al2OH) and 917 cm-1 (AlMgOH) reflect partial substitution of octahedral Al by Mg the smectite clay. So simultaneous presence of the 3617 cm-1, 917 cm-1 and 985 cm-1 bands indicates that the smectite is dioctahedral. The band around 794 cm-1 is due to the presence of illite as well. These results are in agreement with those of the XRD. They confirm the presence of smectite, kaolinite and illite in the studied clay [33].”
Figure 3. FTIR spectra of raw clay RC.
- in 3.4, "...at 110 °C a conversion equal to 98.3% in the first cycle was increased to 81% in the fourth cycle." Why was the increase to 81%.
Thank you for pointing out this mistake, instead of “decreased”, it is “increased”, so we have corrected it in the manuscript
- In this study, the qualitative detection of the product glycerol carbonate is essential, the authors used a mixture to do NMR analysis is not very accurate, it is recommended that glycerol carbonate is separated out for separate NMR analysis or mass spectrometry is better.
Thank you for the recommendation. This study is mainly focused on the activity of the catalyst and the kinetics of the transcarbonation reaction, but we agree with the reviewer that the qualitative analysis of the reagents and products is key. Therefore, we performed 1H NMR analyses both on the target compound as a pure compound or standard (glycerol carbonate) as well as in an sample at mid conversion where all compounds are present in appreciable concentrations. Moreover, we have used ion exclusion HPLC analyses with pure standards, finding that the elution times were exactly those found in reaction samples. In this new version of the manuscript, we have tried to make clearer the figures were they appear, as well as we have extended the discussion. The reviewer can find below the 1H NMR spectra of both pure glycerol carbonate and a reaction sample, as well as a sample ion exclusion HPLC chromatogram.
Figure 6: Expanded region (3.2–5.3 ppm) of 1H NMR spectra in D2O; of a reaction sample containing the mixture Gly, EC, GC and EG
Figure 7: Expanded region (3.2–5.3 ppm) of 1H NMR spectra in D2O of pure glycerol carbonate (GC).
Figure 8. Chromatogram of the reaction between EC and Gly including all reaction species, the internal standard (citric acid) and minor by-products A1 and A2.
- The citation of [37] does not appear in the article. " ...of which they are observed according to the literature at 4.3 ppm, 3.69 ppm and 3.68 ppm, [13-35]." Many irrelevant literature is cited here and the authors are advised to double check.
Thank you for the observation; we have tried our best to double check the citations, in fact, the citation 37 does not appear and I have replaced it with another article which justifies the NMR analysis of glycerol carbonate in a mixture of glycerol and glycerol carbonate, this new citation serves as a reference in the section 3.2. Qualitative and quantitative analysis of reactions. As for the other references, after double-checking, we appreciate that they are adequate to illustrate each point.
[37] Kaur, A.; Prakash , R. ; Ali, A. 1H NMR-assisted quantification of glycerol carbonate in the mixture of glycerol and glycerol carbonate. Talanta. 2018, 178, 1001–1005.

Round 2
Reviewer 3 Report
"Figure 7: Expanded region (3.2–5.3 ppm) of 1H NMR spectra in D2O of pure glycerol carbonate (GC)."what is the solvent for NMR?D2O?or DMSO?
Accept after minor revision
Author Response
Comment
"Figure 7: Expanded region (3.2–5.3 ppm) of 1H NMR spectra in D2O of pure glycerol carbonate (GC)."what is the solvent for NMR?D2O?or DMSO?
We used DMSO to ensure full dissolution of all species. Sorry for the mistake. Now it is corrected in the new version of the manuscript.